# Distribution Learning of a Random Spatial Field with a Location-Unaware Mobile Sensor

**Meera Pai and Animesh Kumar**
Electrical Engineering
Indian Institute of Technology Bombay
Mumbai 400076 India
`meeravpai,animesh@ee.iitb.ac.in`

## Abstract

Measurement of spatial fields is of interest in environment monitoring. Recently mobile sensing has been proposed for spatial field reconstruction, which requires a smaller number of sensors when compared to the traditional paradigm of sensing with static sensors. A challenge in mobile sensing is to overcome the location uncertainty of its sensors. While GPS or other localization methods can reduce this uncertainty, we address a more fundamental question: *can a location-unaware mobile sensor, recording samples on a directed non-uniform random walk, learn the statistical distribution (as a function of space) of an underlying random process (spatial field)?* The answer is in the affirmative for Lipschitz continuous fields, where the accuracy of our distribution-learning method increases with the number of observed field samples (sampling rate). To validate our distribution-learning method, we have created a dataset with 43 experimental trials by measuring sound-level along a fixed path using a location-unaware mobile sound-level meter.

## 1   Introduction

Learning the statistical distribution of physical fields from observed values is a fundamental task in applications like environmental monitoring and pollution control. Consider a spatio-temporal process $X(s,t)$ along a path, such as in a residential neighborhood or a city boulevard, where $s$ denotes the location and $t$ is the time. It is of interest to the learn the statistical distribution of $X(s,t)$ at any point $s$ along the path for environment monitoring. Motivated by this application, the distribution-learning of a Lipschitz continuous spatial field at all locations from spatial samples of its realizations is studied.

In classical environment monitoring done by agencies such as the EPA (http://epa.gov), the sensing locations are assumed to be *known*. This is especially true when there is a dedicated *fixed* sensing location with associated equipment. Recently, mobile-sensing has been proposed as a way to increase the spatial sampling density and reduce the cost of hardware Unnikrishnan and Vetterli [2013]. A key challenge in mobile-sensing is to ascertain the exact location of sampling and it is of interest to work with location-unaware sensing methods Kumar [2017]. While it is possible to use GPS or wireless localization methods to estimate the location, it has energy and hardware overhead Che et al. [2009], Hu and Evans [2004]. We have a more fundamental question: *can recently discovered location-unaware sensing methods be used to learn the statistical distribution of $X(s,t)$ as a function of $s$?* The answer is yes, and analytical and experimental results along this theme will be presented.

Let $X(s,t)$ be a spatial field where $s \in \mathcal{P}$ denotes the location and $t \in \mathbb{R}$ denotes time. The path $\mathcal{P}$ is known, and it can be an open path or a loop. The set $\mathcal{P}$ represents the finite path over which the distribution of $X(s,t)$ has to be learned. It is assumed that $|X(s,t)| \leq b$ everywhere for a finite $b > 0$ and the field is Lipschitz continuous; that is, $|X(s,t) - X(s',t)| \leq \alpha|s - s'|$ for some $\alpha > 0$. The unknown sampling locations are modeled using an *unknown renewal process* (directed non-uniform

random walk) as in the related literature Kumar [2017]. The sampling locations are $S_1, S_2, \ldots, S_M$ along the path $\mathcal{P}$, where $M$ is obtained from the stopping condition $S_M \leq 1, S_{M+1} > 1$. A renewal process implies that $\theta_1 := S_1$, $\theta_2 := S_2 - S_1, \ldots$ are independent and identically distributed. In our setup, the distribution of $\theta$ is not known. This model is useful when there is jitter in mobile sensor's speed or when the sensing time-intervals are programmed to be on a renewal process. The mobile-sensing experiment for distribution-learning is designed around $N$ independent trials. It is assumed that $N$ mobile-sensing experiments, with statistically independent sampling locations between the experiments, are conducted on the path $\mathcal{P}$. Using these location-unaware samples, it is of interest to learn the statistical distribution of $X(s,t)$ for any point $s \in \mathcal{P}$.

Our main results are as follows:

1. Using the classical Glivenko-Cantelli estimate, a distribution-learning method for $X(s,t), s \in \mathcal{P}$ is presented, where the maximum pointwise error between the cumulative distribution function (CDF) of $X(s,t)$ and its estimate decreases as $\mathcal{O}\left(1/(n\varepsilon^2)\right) + \mathcal{O}(\varepsilon)$. Here $\varepsilon > 0$ is a parameter of choice and $n$ is the average number of samples. This result holds in the limit when $N \to \infty$.

2. We have conducted mobile sensing experiments with a sound-level meter. The implications of our distribution-learning method on this custom dataset will be explored, and comparisons between distribution-learning with a fixed and a mobile sound-level meter will be presented.

To apply our distribution-learning method we have measured sound-level along a closed path in multiple experiments. Using a portable sound-level meter, which is location-unaware, a dataset with $N = 43$ trials has been created for the application of the proposed distribution-learning method.

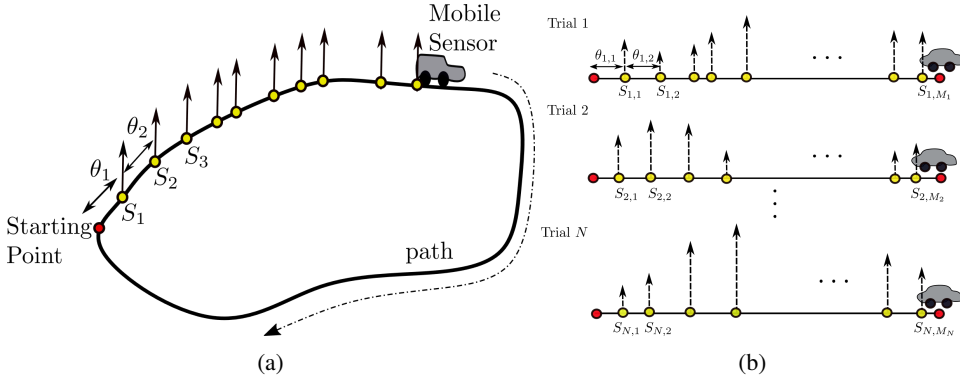

Figure 1: (a) The figure shows a mobile sensor moving along a fixed 1-D path. The field samples are obtained at unknown locations $S_1, S_2, \ldots, S_M$. (b) $N$ trials are carried on $N$ different days. Spatial field values are recorded at unknown locations $S_{i,1}, S_{i,2}, \ldots, S_{i,M_i}$ on trial $i$. The number of samples recorded during the $i$th trial is denoted by $M_i$.

*State of the art:* Classical sampling and distributed sampling have been addressed with fixed sampling locations, where the location of sensor is known A. J. Jerri [1977], Marco et al., Kumar et al. [2011, 2010]. A systematic analysis of spatial sensing with mobile sensor has been studied in Unnikrishnan and Vetterli [2013]. Sensing of temporally fixed parametric spatial fields with location-unaware mobile sensor was first addressed in Kumar [2016]. With location-unaware sampling, interpolation methods for polynomial shapes has been reported in Pacholska et al. [2017]. With location-unawareness, an algorithm for spatial mapping is presented in Elhami et al. [2018]. Mobile sampling is also studied for crowdsensing application; Morselli et al. [2018] compared environmental monitoring using a fixed grid of sensors and sensors attached to vehicles. Use of vehicular sensor networks for environmental monitoring has been studied in Atakan [2014], Wang and Chen [2017].

## 2 Sensing model and spatial field properties

In this section, modeling assumptions made on the spatial field and the location-unaware mobile sensor are presented. First, spatial field properties are discussed.

Let $\mathcal{P}$ be a bounded-length path and $s \in \mathcal{P}$ be a point on it. Let $t$ be time. The spatial field of interest is $X(s,t), s \in \mathcal{P}, t \in \mathbb{R}$. The distribution of $X(s,t)$ as a function of $s \in \mathcal{P}$ has to be 'learned' and it will be termed as the *distribution-learning* problem in this work. The field $X(s,t)$ may be non-stationary as a function of $s \in \mathcal{P}$ which makes the distribution-learning problem non-trivial. It is assumed that $|X(s,t)| \leq b$ everywhere and it is Lipschitz continuous in $s$, i.e.,

$$|X(s,t) - X(s',t)| \leq \alpha|s - s'| \text{ for all } s, s' \in \mathcal{P} \text{ and all } t.$$

The boundedness of spatial derivative indicates that nearby points have similar field values.

Without loss of generality, the one-dimensional path will be considered as $\mathcal{P} = [0,1]$. A location-unaware mobile sensor samples the field $X(s,t)$ from $s = 0$ to $s = 1$ at points generated by an unknown renewal process. The sampling points are $S_1, S_2, \ldots, S_M$, while the inter-sample distances are $\theta_1 := S_1, \theta_2 := S_2 - S_1, \ldots, \theta_M := S_M - S_{M-1}$. The variables $\theta_1, \theta_2, \ldots$ are independent and identically distributed positive random variables. For analysis purposes, it will be assumed that

$$0 < \theta \leq \frac{\lambda}{n} \quad \text{and} \quad \mathbb{E}(\theta) = \frac{1}{n}, \tag{1}$$

where $\lambda > 1$ is finite and represents maximum speed of the sensor while the average sampling rate is $n$/meter.

Since $\theta_1, \theta_2, \ldots$ are assumed to be random variables, the number of sample points realized in $[0,1]$ will be random. Let the random variable $M$ be the number of readings taken in each mobile sensing trial in the interval $\mathcal{P} = [0,1]$. The variable $M$ is given by the following stopping rule Durrett [2010]: $\theta_1 + \theta_2 + \ldots + \theta_M \leq 1$ and $\theta_1 + \theta_2 + \ldots + \theta_{M+1} > 1$. As shown in Kumar [2017], the conditional average of $\theta$ conditioned on $M = n$ is approximately $\frac{1}{n}$. Specifically, it is known that

$$\mathbb{E}[M] \leq n + \lambda - 1. \tag{2}$$

The *distribution* of $\theta_1$ and *the values* of $s_1, \ldots, s_M$ are not required for our distribution-learning algorithm, which makes it a universal learning algorithm under the above assumptions. This is one of the simplest location-unaware mobile sensor model that can be used along a path.

The entire mobile-sensing experiment is designed around $N$ independent trials. It is assumed that the field samples

$$\vec{f_1} := [X(S_{1,1}, t_{1,1}), X(S_{1,2}, t_{1,2}), \ldots, X(S_{1,M_1}, t_{1,M_1})]^T;$$
$$\ldots,$$
$$\vec{f_N} := [X(S_{N,1}, t_{N,1}), X(S_{N,2}, t_{N,2}), \ldots, X(S_{N,M_N}, t_{N,M_N})]^T$$

are available. It is assumed that the observed values in different trials are statistically independent. Using these $N$ different trials, it is of interest to learn the distribution of $X(s,t)$ for any point $s \in \mathcal{P}$. The values of sampling locations $S_{i,j}$ are not known. All of these sampling locations are generated by $N$ independent instances of the same renewal process with inter-sample spacing distribution of $\theta$. Thus, the vectors $\vec{f_1}, \ldots, \vec{f_N}$ are statistically independent. (Individually, each vector $\vec{f_i}$ will be dependent; for example, $S_{1,2} = S_{1,1} + \theta_{1,2}$ depends on $S_{1,1}$.)

## 3  Spatial field's distribution-learning algorithm

This section will summarize our distribution-learning method and the analysis results. The values summarized by $\vec{f_1}, \ldots, \vec{f_N}$ are available. The $i$-th trial results in the dataset $\vec{f_i}$ with $M_i$ number of samples. Since the sample locations are unknown, error in learning the field distribution at any given location depends on the error in the estimation of field values for the given location from samples obtained by the mobile sensor. For any $s \in [0,1]$, the task is to learn the distribution of $X(s,t)$. For notational purposes, in a given trial, let $M$ be the number of recorded samples. Let $M_i$ be the number of samples recorded during trial $i$ and let $S_{i,j}$ denote the location of $j$th sample for trial $i$. From trial $i$, let $\hat{X}_i(s)$ be the estimate of field value at the point $s$ (corresponding to the time of the trial $i$). Designing a good estimate for $\hat{X}_i(s)$ is a challenge in the location-unaware sensing setup. For the distribution-learning problem, we define an estimate for $X(s)$ from the $i$-th trial as

$$\hat{X}_i(s) := X(s_{i,\lfloor (M_i-1)s \rfloor+1}, t_{i,\lfloor (M_i-1)s \rfloor+1}). \tag{3}$$

Note that the dependence on $t$ has been dropped in the left-hand side. This simplified notation will be used, since the main error in distribution-learning will be due to the error in location estimate $s$. The distribution is assumed to be calculated over all time. Let

$$F_{X(s)}(x) = \mathbb{P}(X(s) \le x)$$

denote the cumulative distribution function (CDF) of field values at the location $s$, and let $F_{\hat{X}(s)}(x) = \mathbb{P}(\hat{X}(s) \le x)$ be the CDF of its estimate. Let $1(x \in A)$ be the indicator of set $A$. The CDF of $\hat{X}(s)$ can be obtained as the following classical Glivenko-Cantelli limit:

$$F_{\hat{X}(s)}(x) = \lim_{N \to \infty} \frac{1}{N} \sum_{i=1}^{N} 1\left(\hat{X}_i(s) \le x\right) \tag{4}$$

Our first result establishes the error between $F_{\hat{X}(s)}(x)$ to $F_{X(s)}(x)$ under the previously mentioned location-unaware sensing setup. Let $f_{X(s)}(x)$ be the probability density function of $X(s)$. Then,

**Theorem 1.** *Let $\theta_1, \theta_2, \ldots, \theta_M$ be inter-sample intervals generated by an unknown renewal process such that $\mathbb{E}[\theta_1] = \frac{1}{n}$ and $0 < \theta \le \frac{\lambda}{n}$. Let $M$ be the random number of samples recorded during a trial. Then for every $x \in \mathbb{R}$, $s \in [0, 1]$ and for any $\varepsilon > 0$,*

$$|F_{X(s)}(x) - F_{\hat{X}(s)}(x)| \le \varepsilon . \max\left(f_{X(s)}(x)\right) + \frac{\alpha^2}{\varepsilon^2}((n + \lambda - 1)s(1 - s) + C)\frac{\lambda^2}{n^2}. \tag{5}$$

*Proof.* This result establishes the closeness of CDFs of $X(s)$ and $\hat{X}(s)$ for any $s \in [0, 1]$. Using classical result from [Grimmett and Stirzaker [2001], pg. 311], the following result is noted:

$$F_{\hat{X}(s)}(x) \le F_{X(s)}(x + \varepsilon) + \mathbb{P}\left(\left|\hat{X}(s) - X(s)\right| > \varepsilon\right). \tag{6}$$

When $f_{X(s)}(x)$ exists for every $x$, $|F_{X(s)}(x + \varepsilon) - F_{X(s)}(x)| = \mathbb{P}(x < X(s) \le x + \varepsilon) \le \varepsilon . \max\left(f_{X(s)}(x)\right)$.[1] Therefore,

$$|F_{\hat{X}(s)}(x) - F_{X(s)}(x)| \le |F_{X(s)}(x + \varepsilon) - F_{X(s)}(x)| + \mathbb{P}\left(\left|\hat{X}(s) - X(s)\right| > \varepsilon\right) \tag{7}$$

$$\le \varepsilon . \max\left(f_{X(s)}(x)\right) + \mathbb{P}\left(\left|\hat{X}(s) - X(s)\right| > \varepsilon\right). \tag{8}$$

Since the field is assumed to be Lipschitz continuous, so

$$\left|X(s) - \hat{X}(s)\right| \le \alpha \left|S_{\lfloor (M-1)s \rfloor + 1} - s\right|, \tag{9}$$

where $\alpha$ is the Lipschitz constant. Let

$$l(M, s) = \lfloor (M - 1)s \rfloor + 1.$$

Therefore, the mean-squared error (MSE) in the estimation of spatial field values at location $s$ is given by

$$\mathbb{E}\left[\left|X(s) - \hat{X}(s)\right|^2\right] \le \alpha^2 \mathbb{E}\left[\left|S_{l(M,s)} - s\right|^2\right]. \tag{10}$$

From (23) in Appendix A (given in the supplementary document),

$$\mathbb{E}\left[\left|S_{l(M,s)} - s\right|^2\right] \le (\mathbb{E}[M]s(1 - s) + C)\frac{\lambda^2}{n^2}. \tag{11}$$

From (2), (10), and (11) it follows that

$$\mathbb{E}\left[\left|X(s) - \hat{X}(s)\right|^2\right] \le \alpha^2((n + \lambda - 1)s(1 - s) + C)\frac{\lambda^2}{n^2}. \tag{12}$$

By the Chebyshev's inequality and $\hat{X}(s) = X(S_{l(M,s)})$,

$$\mathbb{P}\left(\left|X(s) - X(S_{l(M,s)})\right| > \varepsilon\right) \leq \frac{1}{\varepsilon^2}\mathbb{E}\left[\left|X(s) - \hat{X}(s)\right|^2\right] \tag{13}$$

$$\leq \frac{\alpha^2}{\varepsilon^2}((n + \lambda - 1)s(1 - s) + C)\frac{\lambda^2}{n^2}. \tag{14}$$

Thus from (8) and (14),

$$|F_{X(s)}(x) - F_{\hat{X}(s)}(x)| \leq \varepsilon.\max\left(f_{X(s)}(x)\right) + \frac{\alpha^2}{\varepsilon^2}((n + \lambda - 1)s(1 - s) + C)\frac{\lambda^2}{n^2}. \tag{15}$$

The second term in the upper bound is of the order $\mathcal{O}(\frac{1}{n\varepsilon^2})$ while the first term is of the order $\mathcal{O}(\varepsilon)$. Therefore, as the sampling rate $n$ tends to infinity, $F_{X(s)}(x)$ converges to $F_{\hat{X}(s)}(x)$. This upper bound depends on $s$ and has a maximum at $s = 1/2$. $\qquad\square$

Similar to the above result, our next theorem obtains a uniform bound on the error between the CDFs of $X(s)$ and $\hat{X}(s)$.

**Theorem 2.** *Let $\theta_1, \theta_2, \ldots, \theta_M$ be inter-sample intervals generated by an unknown renewal process such that $\mathbb{E}[\theta_1] = \frac{1}{n}$ and $0 < \theta \leq \frac{\lambda}{n}$. Let $M$ be the random number of samples recorded during a trial. Then for every $x \in \mathbb{R}, s \in [0, 1]$ and for any $\varepsilon > 0$,*

$$\sup_{s\in[0,1]}\left|F_{\hat{X}(s)}(x) - F_{X(s)}(x)\right| \leq \varepsilon.\max\left(f_{X(s)}(x)\right) + \frac{32}{\beta}\frac{\alpha^2}{\varepsilon^2}(n + \lambda - 1)\frac{\lambda^2}{n^2} \tag{16}$$

*Proof.* From (9),

$$\sup_{s\in[0,1]}\left|X(s) - \hat{X}(s)\right| \leq \alpha \sup_{s\in[0,1]}\left|S_{l(M,s)} - s\right|. \tag{17}$$

For any $\varepsilon > 0$,

$$0 \leq \lim_{n\to\infty}\mathbb{P}\left(\sup_{s\in[0,1]}\left|\hat{X}(s) - X(s)\right| > \varepsilon\right) \leq \lim_{n\to\infty}\mathbb{P}\left(\sup_{s\in[0,1]}\left|S_{l(M,s)} - s\right| > \frac{\varepsilon}{\alpha}\right). \tag{18}$$

Let $\frac{\varepsilon}{\alpha} = \eta$. From (44) in Appendix B (given in the supplementary document),

$$\mathbb{P}\left(\sup_s\left|S_{l(M,s)} - s\right| > \eta\right) \leq \frac{2}{\beta}\frac{16}{\eta^2}\mathbb{E}[M]\frac{\lambda^2}{n^2};$$

where $\beta$ tends to 1 as $n$ tends to infinity. Therefore from (2) and (18),

$$\mathbb{P}\left(\sup_{s\in[0,1]}\left|\hat{X}(s) - X(s)\right| > \varepsilon\right) \leq \frac{32}{\beta}\frac{\alpha^2}{\varepsilon^2}(n + \lambda - 1)\frac{\lambda^2}{n^2} \tag{19}$$

The upper bound in (19) is of $\mathcal{O}(\frac{1}{n})$. This proves that for any $\varepsilon > 0$,

$$\lim_{n\to\infty}\mathbb{P}\left(\sup_{s\in[0,1]}\left|\hat{X}(s) - X(s)\right| > \varepsilon\right) = 0.$$

From (8),

$$\left|F_{\hat{X}(s)}(x) - F_{X(s)}(x)\right| \leq \mathbb{P}\left(\left|X(s) - \hat{X}(s))\right| > \varepsilon\right) + \varepsilon.\max\left(f_{X(s)}(x)\right)$$

$$\leq \mathbb{P}\left(\sup_{s\in[0,1]}\left|X(s) - \hat{X}(s)\right| > \varepsilon\right) + \varepsilon.\max\left(f_{X(s)}(x)\right).$$

The upper bound on the right hand side in (19) is independent of $s$ so,

$$\sup_{s\in[0,1]}\left|F_{\hat{X}(s)}(x) - F_{X(s)}(x)\right| \leq \varepsilon.\max\left(f_{X(s)}(x)\right) + \frac{32}{\beta}\frac{\alpha^2}{\varepsilon^2}(n + \lambda - 1)\frac{\lambda^2}{n^2} \tag{20}$$

This implies that as the sampling rate $n$ tends to infinity, $F_{X(s)}(x)$ converges uniformly over $s \in [0, 1]$ to $F_{\hat{X}(s)}(x)$. $\qquad\square$

In the above result, $\varepsilon$ is a parameter and the upper bound can be minimized over it. The result is left in terms of $\varepsilon$ for future improvements, if any. Simulation results are presented next to validate the above two theorems.

# 4 Simulations for distribution-learning using location-unaware samples

To apply and confirm our distribution-learning method, we consider a synthetic spatiotemporally varying sound-level along a path for simulations. The main goal of these simulations is to verify the accuracy of our distribution-learning method with an increase in the number of samples. The sound-level at location $s \in [0, 1]$ and time $t$ in the simulated signal is $X(s, t)$ where,

$$X(s, t) = \left| 1000 + \sum_{r=1}^{10} A_r(t) \cos(2\pi f_r(t)s) \right|.$$

It is a 10 frequency signal, where the frequencies at each sampling time-instant are generated uniformly at random in the audible frequency range 20 Hz to 20 kHz, and where the amplitudes $A_r(t)$ are generated uniformly in the range $[-180, 180]$; this interval was selected to ensure that the sound-level lies in the usual range of 30-70 dB. Thus, in each trial among a total of $N$, at every sampling instant an independent realization of the sound-level signal is used. Let $t_{i,j}$ be the sampling instants where $i = 1, 2, \ldots, N$ and $j = 1, 2, \ldots, m_i$. Here, $m_i$ is the number of samples collected in trial $i$. The sound-levels for different values of $t_{i,j}$ are independent in our simulations. If $j = l(m_i, s)$, then $X(s, t_{i,j})$ is the true sound-level which is estimated by our algorithm as $\hat{X}(s, t_{i,j}) := X(s_{i,j}, t_{i,j})$ (see (3)). Note that $s_{i,j}$ values are not known in location-unaware sensing; and, these sampling locations are approximated as $\frac{j-1}{m_i}$ for $j = 1, 2, \ldots, m_i$.

The sampling locations are obtained by randomly generated locations $s_{i,1}, s_{i,2}, \ldots s_{i,m_i}$ on trial $i$. These locations are generated by adding independent inter-sample intervals $\theta$ with a Rayleigh distribution having a parameter $\frac{1}{n}\sqrt{\frac{2}{\pi}}$. The mean of $\theta$ is $1/n$. The sound-levels in the simulation are also recorded at $s$ in each trial. These values model the recording of sound-level by a fixed sensor at the point $s$. The empirical CDF of sound-level and their estimates at location $s$ are given by

$$\hat{F}_{X(s)}(x) = \frac{1}{N} \sum_{i=1}^{N} 1\left\{ X(s, t_{i,j}) \leq x \right\} \text{ and } \hat{F}_{\hat{X}(s)}(x) = \frac{1}{N} \sum_{i=1}^{N} 1\left\{ \hat{X}(s, t_{i,j}) \leq x \right\}. \qquad (21)$$

where $1(x \in A)$ denotes the indicator of set $A$. Recall that $j = l(m_i, s)$ for the $i$-th trial as discussed above. Comparisons of CDFs for various values of $n$ and $N$ are shown in Figure 2, where $n$ indicates the sampling rate and $s = 1/2$. As the sampling rate $n$ increases, the number of samples recorded during each trial increases and the error between the estimated CDF of samples obtained by mobile sensor and the actual CDF of samples obtained by the fixed sensor at location $s = 1/2$ reduces. When there is a large number of trials, the error in the estimation of empirical CDFs reduces further. Thus, the simulation results validate our distribution-learning method with location-unaware samples.

# 5 Experiments for sound-level estimation along a path

Sound-level is measured along the path shown in the map in Figure 3 using a sound-level meter. It is carried along the path from the starting point 1 along the path back to point 1. Sound level meter by BAFX products (Model no: BAFX3608) is used for this purpose. Specifications of the sound-level meter are given in Table1. It is not equipped with GPS or any other localization tool.

Table 1: Specifications of Sound Level Meter

| Range: 30-130dB | Sampling Rate: 1 per sec | Memory: 4700 readings | Accuracy: $\pm$ 1.5 dB |
| --- | --- | --- | --- |

*Datasets:* We have created two different datasets by measuring sound-level along the path shown in Figure 3. For the first data set denoted by Dataset1, the path is traversed with a sound-level meter. It begins recording data at the starting point and continues collecting data along the entire path. This acts as a location-unaware mobile sound-level meter. A static sound-level meter is used to measure sound-level at specific locations marked in the map in Figure 3 with numbers one to nine, during each trial. This acts as a fixed sensor as the field is measured at known locations. We have performed 43 trials along the same path in Figure 3. For the second dataset denoted by Dataset2, the path

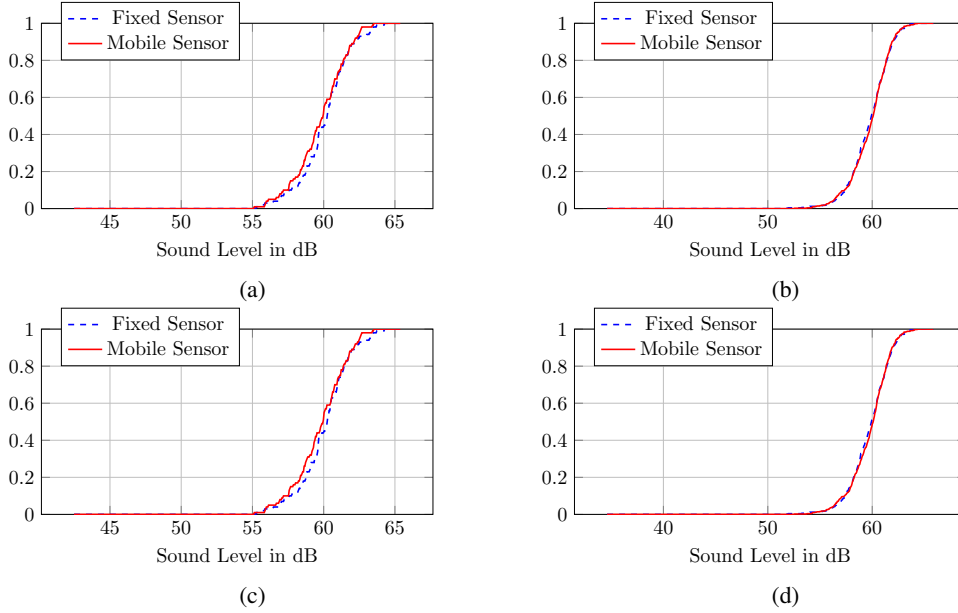

Figure 2: Empirical CDF of simulated sound-level at the location $s = 0.5$ where $n$ is the sampling rate and $N$ is the number of trials: (a) $n = 100$ and $N = 100$; (b) $n = 100$ and $N = 500$; (c) $n = 1000$ and $N = 100$; and, (d) $n = 1000$ and $N = 500$.

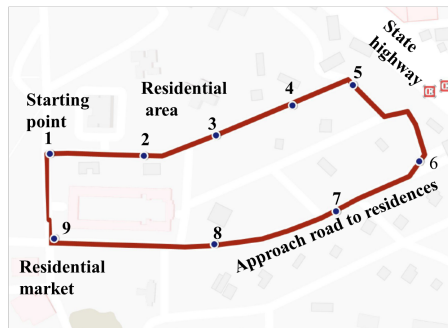

Figure 3: Path along which sound-level is recorded. The locations marked in the map with numbers are used for measurement using a fixed sensor

is traversed using the sound-level meter while cycling, where sampling rate in space is *lower* as compared to walking. We have performed 43 trials in this case as well along the same path in Figure 3. Since the sound-level meter records samples at the rate of 1 sample per second, the spatial sampling rate for Dataset2 is smaller than the spatial sampling rate for Dataset1. We have also emulated a fixed station at location 9 in Figure 3 using a static sensor for 10 minutes.

For experimentation, the path in Figure 3 was chosen as there is a large variation in the sound-level along the path. The residential area is expected to be quiet compared to the region near the state highway and residential market. The box plot for Dataset1 is illustrated in Figure 4. A box plot displays information about the range, median, and quartiles of the data. From Figure 4, the dynamic range of sound-level along the path is observed. The average sound-level variation is 20 dB (ratio of 100) while the dynamic range exceeds 30 dB (ratio of 1000). The main aim is to apply the distribution-learning method on experimental data, and compare the agreement of learned distributions between a mobile sensor and a fixed sensor. The empirical distribution of sound-level obtained from the mobile sound-level meter defined in (21) and the empirical distribution of sound-level obtained from the fixed sensor defined in (21) that measures sound-level at locations marked with numbers 1-9 in Figure 3 is compared. Figure 5a shows the comparison of empirical CDFs of experimental data from Dataset1 at location 5 in Figure 3. The error in the empirical distributions computed using samples from the

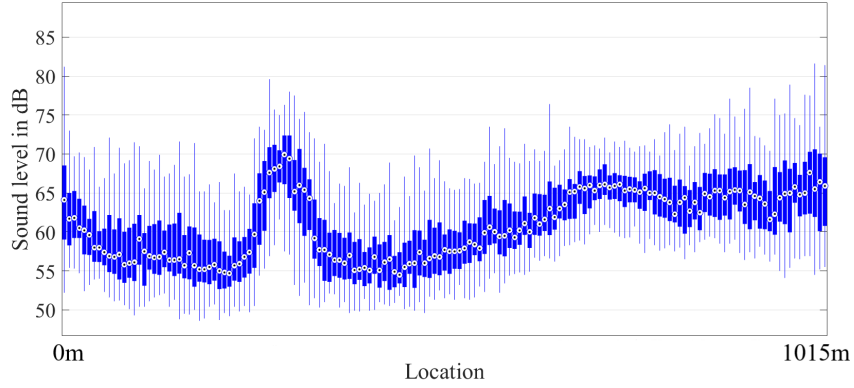

Figure 4: Box plot for samples obtained from the mobile sound level device in Dataset1 along the path in Figure 3 of length 1015 meter is illustrated.

fixed sensor and the mobile sensor in Dataset1 is small as shown in Figure 5a. This shows that the sound-level distribution at any location on a path can be learned using location-unaware samples.

To check the distribution-learning method at two different sampling rates of the mobile sound-level meter, the empirical CDF of sound-level defined by (21) (at location 9 in Figure 3) using a fixed sensor and empirical CDF of sound-level obtained by mobile sensors defined by (21) are compared. This comparison is done at two different sampling rates, obtained from Dataset1 and Dataset2. The CDFs are plotted in Figure 5. From Figure 5(b) and (c) the accuracy in learning the distribution is better for Dataset1 (higher spatial sampling rate) as compared to Dataset2 (lower spatial sampling rate). The accuracy of the distribution-learning method increases with spatial sampling rate. The decrease in maximum pointwise error in learned CDF with $n$ is also shown in Theorems 1 and 2.

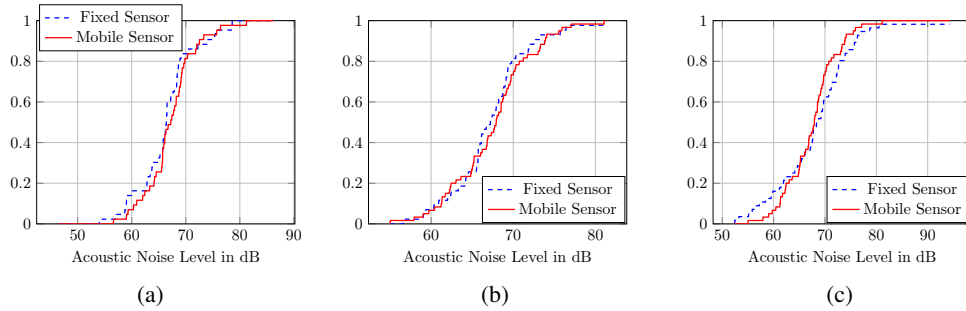

Figure 5: (*a*) Comparison of empirical CDF of sound-level at location 5 in Figure 3, obtained by the fixed sensor and by experimentation at location 9 in Figure 3, for two different sampling rates of mobile sensor: (*b*) fixed sensor versus mobile sensor for Dataset1 (Higher spatial sampling rate) (*c*) fixed sensor versus mobile sensor for Dataset2 (Lower spatial sampling rate)

## 6   Conclusions

In this work, we proposed a data-driven method for learning the statistical distribution of a Lipschitz continuous spatial field along a path. The samples used were obtained at unknown-locations generated by an unknown renewal process. The accuracy of the proposed distribution-learning method increases with the spatial sampling rate of the mobile sensor. Simulation and experimental results support this claim. A method to learn the variation of distribution with time needs be developed if the field is temporally varying in nature. The field was assumed to be one dimensional and a single mobile sensor was used to sample. Use of multiple location-unaware mobile sensors for sampling 2-D fields can be studied in the future.

## Footnotes

[1] In case $f_{X(s)}(x)$ does not exist for every $x$, since $F_{X(s)}(x)$ is a continuous function for every $\varepsilon > 0$ there exists a $\delta(\varepsilon) > 0$ such that $|F_{X(s)}(x + \varepsilon) - F_{X(s)}(x)| \le \delta(\varepsilon)$. As $\varepsilon$ tends to zero $\delta(\varepsilon)$ tends to zero.

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
