[Supplementary Material]

# Appendix: Distribution Learning of a Random Spatial Field with a Location-Unaware Mobile Sensor

**Meera Pai and Animesh Kumar**
Electrical Engineering
Indian Institute of Technology Bombay
Mumbai 400076 India
meeravpai,animesh@ee.iitb.ac.in

## A

The following results are proved in Appendix A in Kumar [2017].

$$\mathbb{E}[\theta|M=m] = \frac{1}{m} - \frac{\mathbb{E}[R_M^2|M=m]}{m} \tag{1}$$

since $R_M^2 \leq \frac{\lambda^2}{n^2}$, for large sampling rate, the second term in the above equation is negligible. It is shown(in Appendix A in Kumar [2017]) that

$$ma_m + m(m-1)b_m = \mathbb{E}[R_M^2|M=m]$$

or

$$b_m = \frac{1}{m(m-1)}\left(-ma_m + \mathbb{E}[R_M^2|M=m]\right) \tag{2}$$

where $a_m = \mathbb{E}\left[\left(\theta_1 - \frac{1}{m}\right)^2 \Big| M=m\right]$ and $b_m = \mathbb{E}\left[\left(\theta_1 - \frac{1}{m}\right)\left(\theta_2 - \frac{1}{m}\right)\Big| M=m\right].$

The mean squared error between location $s$ and $S_{\lfloor(M-1)s\rfloor+1}$ conditioned on M=m is

$$\mathbb{E}\left[\left|S_{l(M,s)} - s\right|^2 \Big| M=m\right] \tag{3}$$

$$= \mathbb{E}\left[\left|S_{l(M,s)} - \frac{l(M,s)}{M} + \frac{l(M,s)}{M} - s\right|^2 \Big| M=m\right] \tag{4}$$

$$= \mathbb{E}\left[\left|S_{l(M,s)} - \frac{l(M,s)}{m}\right|^2 \Big| M=m\right] + \mathbb{E}\left[\left|\frac{l(M,s)}{m} - s\right|^2 \Big| M=m\right] \tag{5}$$

$$+ 2\mathbb{E}\left[\left(S_{l(M,s)} - \frac{l(M,s)}{m}\right)\left(\frac{l(M,s)}{m} - s\right)\Big| M=m\right]. \tag{6}$$

The term $\frac{l(M,s)}{m} - s$ in the above equation can be simplified as, $\frac{l(m,s)}{m} - s = \frac{\lfloor(m-1)s+1\rfloor}{m} - s \leq \frac{1}{m}$. Therefore,

$$\mathbb{E}\left[\left|S_{l(M,s)} - s\right|^2 \Big| M=m\right] \leq \mathbb{E}\left[\left|S_{l(M,s)} - \frac{l(M,s)}{m}\right|^2 \Big| M=m\right]$$
$$+ \frac{1}{m^2} + 2\frac{1}{m}\mathbb{E}\left[S_{l(M,s)} - \frac{l(M,s)}{m}\Big| M=m\right]. \tag{7}$$

The first term in the Right hand side (RHS) of (7) is

$$\mathbb{E}\left[\left|S_{l(M,s)} - \frac{l(M,s)}{m}\right|^2 \bigg| M = m\right]$$

$$= \mathbb{E}\left[\left(\sum_{i=1}^{l(M,s)} \left(\theta_i - \frac{1}{m}\right)\right)^2 \bigg| M = m\right]$$

$$= \mathbb{E}\left[\left(\sum_{i=1}^{l(M,s)} \sum_{j=1}^{l(M,s)} \left(\theta_i - \frac{1}{m}\right)\left(\theta_j - \frac{1}{m}\right)\right) \bigg| M = m\right]$$

$$= (l(m,s))\mathbb{E}\left[\left(\theta_i - \frac{1}{m}\right)^2 \bigg| M = m\right]$$

$$+ (l(m,s))(l(m,s)-1)\mathbb{E}\left[\left(\theta_i - \frac{1}{m}\right)\left(\theta_j - \frac{1}{m}\right) \bigg| M = m\right] \text{ where } i \neq j$$

$$\leq ((m-1)s+1)a_m + ((m-1)s+1)((m-1)s)b_m$$

where $l(m,s) = \lfloor (m-1)s \rfloor + 1$.

Substituting for $b_m$ from equation (2) we get,

$$\mathbb{E}\left[\left|S_{l(M,s)} - \frac{l(M,s)}{m}\right|^2 \bigg| M = m\right] \tag{8}$$

$$\leq ((m-1)s+1)a_m + \frac{((m-1)s+1)((m-1)s)}{m(m-1)}\left(-ma_m + \frac{\lambda^2}{n^2}\right) \tag{9}$$

$$= ((m-1)s+1)a_m - ((m-1)s^2+s)a_m + \left(\frac{(m-1)s^2+s}{m}\right)\frac{\lambda^2}{n^2} \tag{10}$$

$$= (m-1)s(1-s)a_m + (1-s)a_m + \left(\frac{(m-1)s^2+s}{m}\right)\frac{\lambda^2}{n^2}. \tag{11}$$

The above equation can be simplified by substituting for $a_m$ defined as

$$a_m = \mathbb{E}\left[\left(\theta - \frac{1}{m}\right)^2 \bigg| M = m\right] = \mathbb{E}\left[\theta^2 - 2\frac{\theta}{m} + \frac{1}{m^2} \bigg| M = m\right].$$

Since $\mathbb{E}[\theta|M = m] = \frac{1}{m}$ from (1) and $\mathbb{E}[\theta^2] \leq \frac{\lambda^2}{n^2}$,

$$a_m \leq \frac{\lambda^2}{n^2} - \frac{1}{m^2}.$$

Substituting the above upper bound in (11) we get

$$\mathbb{E}\left[\left|S_{l(M,s)} - \frac{l(M,s)}{m}\right|^2 \bigg| M = m\right] \tag{12}$$

$$\leq (m-1)s(1-s)\left(\frac{\lambda^2}{n^2} - \frac{1}{m^2}\right) + (1-s)\left(\frac{\lambda^2}{n^2} - \frac{1}{m^2}\right) + \left(\frac{(m-1)s^2+s}{m}\right)\frac{\lambda^2}{n^2}. \tag{13}$$

$$\tag{14}$$

By replacing $(m-1)$ by $m$ and rearranging the terms in above equation we can write,

$$\mathbb{E}\left[\left|\left|S_{l(M,s)} - \frac{l(M,s)}{m}\right|\right|^2 \middle| M = m\right] \tag{15}$$

$$\leq \left(ms(1-s) + 1 - s + s^2 + \frac{s}{m}\right)\frac{\lambda^2}{n^2} - (ms(1-s) + 1 - s)\frac{1}{m^2}. \tag{16}$$

$$\tag{17}$$

The third term in the RHS of equation (7) can be simplified as

$$\mathbb{E}\left[S_{l(M,s)} - \frac{l(M,s)}{m}\middle| M = m\right] = \mathbb{E}\left[\sum_{i=1}^{l(M,s)} \theta_i - \frac{l(M,s)}{m}\middle| M = m\right] \tag{18}$$

$$= l(m,s)\mathbb{E}[\theta|M=m] - \frac{l(m,s)}{m} = 0. \tag{19}$$

Therefore, putting together equations (7) and (16), (19) we can write,

$$\mathbb{E}\left[\left|S_{l(M,s)} - s\right|^2\right] \tag{20}$$

$$\leq \left(\mathbb{E}[M]s(1-s) + 1 - s + s^2 + \frac{s}{\mathbb{E}[M]}\right)\frac{\lambda^2}{n^2} - (\mathbb{E}[M]s(1-s) + 1 - s)\frac{1}{\mathbb{E}[M]^2} + \frac{1}{\mathbb{E}[M]^2} \tag{21}$$

$$\leq \left(\mathbb{E}[M]s(1-s) + 1 - s + s^2 + \frac{s}{\mathbb{E}[M]}\right)\frac{\lambda^2}{n^2} \tag{22}$$

$$= (\mathbb{E}[M]s(1-s) + C)\frac{\lambda^2}{n^2}, \tag{23}$$

where C is a constant.

## B

The **Firt Symmetrization Lemma**[Pollard [2012]] states that: Let $Z(t) : t \in T$ and $Z'(t) : t \in T$ be independent stochastic processes sharing an index set $T$. Suppose there exist constants $\beta > 0$ and $\gamma > 0$ such that $\mathbb{P}\{|Z'(t)| \leq \gamma\} \geq \beta$ for every $t \in T$, then

$$\mathbb{P}\{\sup_t |Z(t)| \geq \varepsilon\} \leq \frac{1}{\beta}\mathbb{P}\{\sup_t |Z(t) - Z'(t)| > \varepsilon - \gamma\} \tag{24}$$

Let us define $Z(s) = S_{l(M,s)} - s$. Let $Z'(s)$ be independent of $Z(s)$ sharing the same index set $s \in [0,1]$, generated by a different set of sampling locations for the same number of total samples i.e $Z'(s) = S'_{l(M,s)} - s$.

Using the upper bound in (20) in Appendix A,

$$\mathbb{P}\left(\left|S'_{l(M,s)} - s\right| \leq \gamma\right) = 1 - \mathbb{P}\left(\left|S'_{l(M,s)} - s\right| > \gamma\right)$$

$$\geq 1 - \frac{\mathbb{E}\left[\left|S'_{l(M,s)} - s\right|^2\right]}{\gamma^2}$$

$$\geq 1 - \left(\mathbb{E}[M]s(1-s) + 1 - s + s^2 + \frac{s}{\mathbb{E}[M]}\right)\frac{\lambda^2}{n^2\gamma^2}.$$

i.e. $\mathbb{P}\{|Z'(s)| \leq \gamma\} \geq \beta$ where

$$\beta = 1 - \left(\mathbb{E}[M]s(1-s) + 1 - s + s^2 + \frac{s}{\mathbb{E}[M]}\right)\frac{\lambda^2}{n^2\gamma^2}. \tag{25}$$

$\beta$ goes to 1 as $n \to \infty$.

Using the first symmetrization lemma we can write,

$$\mathbb{P}\{\sup_s \left|S_{l(M,s)} - s\right| > \varepsilon\} \le \frac{1}{\beta}\mathbb{P}\left\{ \sup_s \left|S_{l(M,s)} - s - S'_{l(M,s)} + s\right| > \varepsilon - \gamma\right\}.$$

$$= \frac{1}{\beta}\mathbb{P}\left\{ \sup_s \left|S_{l(M,s)} - S'_{l(M,s)}\right| > \varepsilon - \gamma\right\}$$

$$= \frac{1}{\beta}\mathbb{P}\left\{ \sup_s \left|\sum_{i=1}^{l(M,s)} (\theta_i - \theta'_i)\right| > \varepsilon - \gamma\right\}.$$

Taking $\gamma = \frac{\varepsilon}{2}$ in the above equation we get,

$$\mathbb{P}\{\sup_s \left|S_{l(M,s)} - s\right| > \varepsilon\} \le \frac{1}{\beta}\mathbb{P}\left\{ \sup_s \left|\sum_{i=1}^{l(M,s)} (\theta_i - \theta'_i)\right| > \frac{\varepsilon}{2}\right\} \tag{26}$$

Using the second symmetrization lemma[Pollard [2012]],

$$\mathbb{P}\{\sup_s \left|S_{l(M,s)} - s\right| > \varepsilon\} \le \frac{2}{\beta}\mathbb{P}\left\{ \sup_s \left|\sum_{i=1}^{l(M,s)} \sigma_i\theta_i\right| > \frac{\varepsilon}{4}\right\}.$$

where $\sigma_1, \sigma_2, \sigma_3, \dots$ are i.i.d Rademacher random variables that are also independent of $\theta_i$, $\theta'_i$ and $\mathbb{P}\{\sigma_i = +1\} = \mathbb{P}\{\sigma_i = -1\} = \frac{1}{2}$.

From Figure 1 we can see that, $\left|\sum_{i=1}^{l(M,s)} \sigma_i\theta_i\right|$ reaches it maximum value when $s$ becomes $\frac{k}{M-1}$. Therefore supremum over $s$ can be written as maximum over $\frac{k}{M-1}$, i.e.

Figure 1: $\sum_{i=1}^{\lfloor (M-1)s\rfloor+1} \sigma_i\theta_i$ plotted against $s$. *We can see that depending on the sign of $\sigma_i$, $\theta_i$ either gets added or subtracted from $\sum_1^{i-1} \sigma_k\theta_k$*

$$\mathbb{P}\{\sup_s \left|S_{l(M,s)} - s\right| > \varepsilon\} \le \frac{2}{\beta}\mathbb{P}\left\{ \max_k \left|\sum_{i=1}^{k+1} \sigma_i\theta_i\right| > \frac{\varepsilon}{4}\right\} \text{ where } k = 0, 1, \dots (M-1).$$

Using Chebyshev's inequality we get,

$$\mathbb{P}\{\sup_s \left|S_{l(M,s)} - s\right| > \varepsilon\} \le \frac{2}{\beta}\frac{16}{\varepsilon^2}\mathbb{E}\left[ \max_k \left|\sum_{i=1}^{k+1} \sigma_i\theta_i\right|^2\right]. \tag{27}$$

Figure 2: plot of $\exp(x)$ and $(e-1)x + 1$ for $x$ between 0 and 1

From Jensen's inequality,

$$\exp\left(s\mathbb{E}\left[\max_k\left|\sum_{i=1}^{k+1}\sigma_i\theta_i\right|^2\Bigg|M=m\right]\right) \tag{28}$$

$$\leq \mathbb{E}\left[\exp\left(s\max_k\left|\sum_{i=1}^{k+1}\sigma_i\theta_i\right|^2\right)\Bigg|M=m\right] \tag{29}$$

$$= \mathbb{E}\left[\max_k\exp\left(s\left|\sum_{i=1}^{k+1}\sigma_i\theta_i\right|^2\right)\Bigg|M=m\right] \tag{30}$$

$$= \mathbb{E}\left[\max_k\exp\left(s\sum_{i=1}^{k+1}\sum_{j=1}^{k+1}\sigma_i\theta_i\sigma_j\theta_j\right)\Bigg|M=m\right] \tag{31}$$

$$= \mathbb{E}\left[\max_k\exp\left(s\left(\sum_{i=1}^{k+1}\sigma_i^2\theta_i^2 + \sum_{i=1}^{k+1}\sum_{j=1}^{k+1}\sigma_i\theta_i\sigma_j\theta_j\right)\right)\Bigg|M=m\right] \text{ where } i \neq j \tag{32}$$

$$\leq \mathbb{E}\left[\max_k\exp\left(s\left(\sum_{i=1}^{k+1}\frac{\lambda^2}{n^2} + \sum_{i=1}^{k+1}\sum_{j=1}^{k+1}\sigma_i\theta_i\sigma_j\theta_j\right)\right)\Bigg|M=m\right] \tag{33}$$

$$= \mathbb{E}\left[\max_k\exp\left(s\left((k+1)\frac{\lambda^2}{n^2} + \sum_{i=1}^{k+1}\sum_{j=1}^{k+1}\sigma_i\theta_i\sigma_j\theta_j\right)\right)\Bigg|M=m\right] \tag{34}$$

$$\leq \mathbb{E}\left[\max_k\exp\left(sm\frac{\lambda^2}{n^2}\right)\exp\left(s\sum_{i=1}^{k+1}\sum_{j=1}^{k+1}\sigma_i\theta_i\sigma_j\theta_j\right)\Bigg|M=m\right] \tag{35}$$

The maximum path length is one therefore, $\left|\sum_{i=1}^{k+1}\sigma_i\theta_i\right|^2 \leq 1$. This implies that, $\sum_{i=1}^{k+1}\sum_{j=1}^{k+1}\sigma_i\theta_i\sigma_j\theta_j$ where $i \neq j$ is less than 1. From figure 2 we can see that for $0 \leq x \leq 1$, $\exp(x) \leq (e-1)x + 1$. So, by replacing $\exp\left(s\sum_{i=1}^{k+1}\sum_{j=1}^{k+1}\sigma_i\theta_i\sigma_j\theta_j\right)$ with

$(e-1)s\sum_{i=1}^{k+1}\sum_{j=1}^{k+1}\sigma_i\theta_i\sigma_j\theta_j + 1$ we get

$$\exp\left(s\mathbb{E}\left[\max_k\left|\sum_{i=1}^{k+1}\sigma_i\theta_i\right|^2\middle| M=m\right]\right) \tag{36}$$

$$\leq \mathbb{E}\left[\exp\left(sm\frac{\lambda^2}{n^2}\right)\max_k\left((e-1)s\sum_{i=1}^{k+1}\sum_{j=1}^{k+1}\sigma_i\theta_i\sigma_j\theta_j + 1\right)\middle| M=m\right] \tag{37}$$

$$= \mathbb{E}\left[\exp\left(sm\frac{\lambda^2}{n^2}\right)\left(1+\max_k(e-1)s\sum_{i=1}^{k+1}\sum_{j=1}^{k+1}\sigma_i\theta_i\sigma_j\theta_j\right)\middle| M=m\right] \tag{38}$$

$$\tag{39}$$

Since $\max_k(e-1)s\sum_{i=1}^{k+1}\sum_{j=1}^{k+1}\sigma_i\theta_i\sigma_j\theta_j \leq \sum_{k=0}^{m-1}(e-1)s\sum_{i=1}^{k+1}\sum_{j=1}^{k+1}\sigma_i\theta_i\sigma_j\theta_j$,

$$\exp\left(s\mathbb{E}\left[\max_k\left|\sum_{i=1}^{k+1}\sigma_i\theta_i\right|^2\middle| M=m\right]\right) \tag{40}$$

$$\leq \mathbb{E}\left[\exp\left(sm\frac{\lambda^2}{n^2}\right)+\exp\left(sm\frac{\lambda^2}{n^2}\right)\sum_{k=0}^{m-1}(e-1)s\sum_{i=1}^{k+1}\sum_{j=1}^{k+1}\sigma_i\theta_i\sigma_j\theta_j\middle| M=m\right] \tag{41}$$

$$= \exp\left(sm\frac{\lambda^2}{n^2}\right)+\exp\left(sm\frac{\lambda^2}{n^2}\right)\sum_{k=0}^{m-1}(e-1)s\sum_{i=1}^{k+1}\sum_{j=1}^{k+1}\mathbb{E}[\sigma_i\theta_i\sigma_j\theta_j|M=m] \tag{42}$$

As $\theta$ and $\sigma$ are independent of each other and $\mathbb{E}[\sigma]=0$, $\mathbb{E}[\sigma_i\theta_i\sigma_j\theta_j|M=m]=0$ therefore,

$$\exp\left(s\mathbb{E}\left[\max_k\left|\sum_{i=1}^{k+1}\sigma_i\theta_i\right|^2\middle| M=m\right]\right) \leq \exp\left(sm\frac{\lambda^2}{n^2}\right) \tag{43}$$

Taking natural log on both sides of above equation gives,

$$\mathbb{E}\left[\max_k\left|\sum_{i=1}^{k+1}\sigma_i\theta_i\right|^2\middle| M=m\right] \leq m\frac{\lambda^2}{n^2}. \tag{44}$$

Therefore,

$$\mathbb{E}\left[\max_k\left|\sum_{i=1}^{k+1}\sigma_i\theta_i\right|^2\right] \leq \mathbb{E}[M]\frac{\lambda^2}{n^2}. \tag{45}$$

Substituting the above bound in 27 we get,

$$\mathbb{P}\{\sup_s\left|S_{l(M,s)}-s\right| > \varepsilon\} \leq \frac{2}{\beta}\frac{16}{\varepsilon^2}\mathbb{E}[M]\frac{\lambda^2}{n^2} \tag{46}$$

$$\leq \frac{2}{\beta}\frac{16}{\varepsilon^2}(n+\lambda-1)\frac{\lambda^2}{n^2} \tag{47}$$