[Reviews · NeurIPS 2019]

Reviewer 1



The setting is interesting and the potential contribution here could be significant. Also, in terms of clarity and presentation, the authors have made a very good job, in my opinion. However, although overall this has the potential to be a good paper, at this point I think there are some issues. In particular: 1. In (4), N has to tend to infinity, which may not be practical in many applications of the type considered in the paper. Instead, one would be more interested in an actually finite sample guarantee for the proposed estimator, which would be also implementable. The limiting operation in (4) seems kind of restrictive and thus practical merit of the work limited. 2. The second main result presented, Theorem 2, is quite strange. In (5), with respect to what is the maximization involving the pdf on the right-hand-side? I guess x. However, if this is true, then what happens later in (16)? Something seems not right here, because in (16) one could take the infimum of the right-hand-side, and get a better bound. However, you do not seem to do something like that. Please explain in sufficient detail what is happening. Also, (20) does not seem right to me, because the first term on the right-hand-side depends on s, and it seems that this dependence has been ignored when taking the supremum over s, on both sides of the preceding expression. 3. The experiments presented do not really match the setting studied in the analysis. For example, in lines 155 - 157: The application does not really justify the theoretical analysis. If N is days, then N cannot be something really large right? I mean, for how many days are you going to measure the acoustic field X anyways? In particular, I see that N=43, which is rather small to be supported by the theoretical results, where N tends to infinity. Therefore, based on the above, this paper might not be of sufficient quality for acceptance. However, I am willing change my opinion as long as the responses by the authors are satisfactory, especially regarding Theorem 2, which an important part of the contributions claimed. ===========After Author Response========== I have also read the author response. The fact that the authors state that it might be possible to study the problem in the finite sample regime makes me wonder that the paper might be possible to be strengthened (with a new version), in order to support the particular application stated in the title. However, I still believe that the results are interesting. For now, I will increase to my score to 6.

Reviewer 2



The paper deals with distribution learning for a spatial field in the context of mobile sensors but location-unaware. As far as I understand, the paper resorts to the standard empirical cumulative density function as a tool for learning. The paper seems technically sound and clearly written. The empirical cumulative density is an usual estimate of the distribution. The paper proposes a specific study of its properties in the considered context. To this end the inter-sample intervals are modeled as a renewal process. It could be interesting to explain why this model is a realistic one. I must add that I am not able to check the proof because they are out of the scope of my research. Regarding the experimental study of Section 5, it is correctly described but the analysis ans comment lines 210-211 must be developed.

Reviewer 3



Update: I have read the author response. - I acknowledge that topic is within the general scope of NeurIPS (especially under the signal processing/time series area). I've increased my score from 3 to 4 accordingly, and updated my review to reflect this. - Regarding the metric for measuring accuracy, I meant to suggest it would be useful to be more precise earlier in the paper about what metric is being used, since there are many ways to measure the error between two distributions (\ell_1, \ell_2, TV, KL, ...). - In the experiments, consider reporting the error metric used in the Theorems as well as the upper bounds, in addition to plotting the CDFs. - I agree that C2 is encompassed by your assumptions, but I would also expect that stronger results may be achievable if one makes stronger assumptions (such as C2), and in some applications it may be very reasonable to make a stronger assumption. It's fair to say that this is for future work. My point is that having a concrete motivating application scenario (e.g., monitoring air quality in doors) would make it easier to justify modeling assumptions. --- This paper mathematically formulates the problem of estimating a spatial field using a location-unaware mobile sensor and it proposes an algorithm for distribution learning in this setting. The problem formulation, algorithm, and analysis of the algorithm appear to be novel and original. The main weakness of this paper, and the reason my overall score is "clear reject", is the lack of strong motivation and justification for the problem formulation and assumptions. When is it useful to have an estimate of the distribution of values along a path (i.e., a concrete motivating application)? For that application, what is an appropriate metric to use for measuring the accuracy of a method? Why is it reasonable to only assume that the field is Lipschitz? Why not something stronger (e.g., C^2)? Why is it reasonable to assume that the observations are not corrupted by any noise (so the only randomness is due to uncertainty about the position)? In addition, there were some aspects of the problem formulation that were not clear to me upon reaching the end of Sec 2. - What is the goal? (Distribution learning, but no specific metric or way of measuring performance was described, nor were baselines or fundamental performance limits discussed) - It also wasn't clear that a "path" really means a closed loop with known the starting and ending point always being the same. The example signal simulated in Sec 4 isn't time-varying. Is that intentional? There is some mismatch between the title (which would imply that the aim is to estimate a random field) and the actual setting of the paper (the field is deterministic, the sampling locations are random).

[Author Response · NeurIPS 2019]

We would like to thank all the reviewers for their selfless reviews and astute comments about our work. We sincerely think that these comments will improve the quality of our paper's presentation. We provide a brief point by point reply to the concerns raised, while abridging the reviewers' concerns due to page constraint.

**Reviewer 1:**

*Point 1. about finiteness of $N$:* This is an interesting point. For finite $N$, we note that the empirical CDF is a bounded random variable. So Hoeffding or any sub-Gaussian random variable tail inequality will result in a sharp concentration around the mean (true CDF) for finite $N$. Owing to space and deadline constraints, we will consider this in a future version of our work.

*Point 2. about Theorem 2:* In (5), (16) and (20) the $\max$ is over $x$ and $s$. Therefore, only the second term in the right hand side of (5) is dependent on $s$. In Theorem 2 we take the supremum over $s$ which results in (16) and (20).

*Point 3. about experiments and value of $N$:* In our trials, as noted by the reviewer, $N$ is indeed days. It is indeed shy of infinity. However, we do see "closeness" between the empirical CDFs obtained by fixed sensors readings and mobile sensor readings. This was the point of experiments. We expect this agreement to get better with larger $N$.

**Reviewer 2:**

*Point 1. about renewal process model:* The renewal model is realistic in the following scenarios: (i) if the mobile vehicle is nearly on uniform speed with slight variation in the speed (jitter); and (ii) if the sensing time intervals are programmed to record on a (temporal) renewal process. We will add the above points to clarify the renewal process model used. We believe that inter-sample intervals *can be* modeled as autoregressive process, but it is beyond the scope of our current work.

*Point 2. about experimental study:* Owing to space constraints, we abridged this discussion. We will expound more on it if we get a chance to revise our NeurIPS submission.

*Point 3. about minor details:* We thank the reviewer for suggesting minor changes. We will incorporate them in our final paper.

**Reviewer 3:**

*Points raised in paragraph 2:* In spatial sensing setup (such as in smart cities or IoT or climatology), it is desirable to estimate the distribution of spatial fields along a path or in a region; this is our upcoming application. In distribution learning, error between the estimated and the true distribution is an accuracy metric. (On Lipschitz and $C^2$) We note that a spatial field in $C^2$ is also Lipschitz and therefore our results will extend to the $C^2$ setup; and, we need Lipschitz criterion since smoothness of the field seems necessary when measurement locations have small errors. (On NeurIPS and signal processing) We did submit our work to the signal processing/time series area of NeurIPS. We also note that our work presents a data-driven approach (backed by proofs) to distribution learning, which we believe is a good fit for NeurIPS.

*Points raised on the lack of clarity in problem formulation:* We will be more than happy to rectify any lack of clarity in our paper. While we will give a thorough read, it would benefit us if the reviewer points out the major issue.

*On the goal of our paper and lack of fundamental limits:* This is a great point. Our paper is a first attempt towards distribution learning with location-unaware sensor. It is a new area of interest, which involves contributions to statistical learning theory, since the distribution of sampling location is not known. As a result, benchmarks/fundamental limits are not available yet. We believe our positive result (i.e., distribution can be learned with higher density of samples) will be of interest to the community. We do wish to unravel fundamental limits as we study this area further.

*On the path being closed or a loop:* Please note that the begin and end point of the path need not be the same, though in our experiments they were. We will revise our submission to remove any confusion regarding this issue.

*On the lack of time-variation in simulated signal:* Upon reviewing our paper submission, it does seem that $A_r(s), f_r(s)$ do not vary with time. But, we generated $A_r(s), f_r(s)$ for each renewal process based sampling location $S_{i,j}$ independently ($i$-th trial, $j$-th sampling location). This independent realization of $A_r(S_{i,j})$ and $f_r(S_{i,j})$ is modeling the time-variation. In our experiments, the sampling time interval is $1$ second, during which the spatial acoustic field modifies to a different value. We will *revise* the simulation section to reflect this very important point.

*On mismatch between the title and setting:* We think there is no mismatch since apart from our assumptions on Lipschitz property of $X(s,t)$, we are working with a random process in Theorem 1 and Theorem 2. In experiments, the spatial field changes (is not deterministic or fixed) as our mobile sensor moves. In the simulations as well, the field's frequency and amplitude changes to a random value between successive samples. We request the reviewer to check the same. In other words, the spatial field is random and the sampling locations are random too.

[Meta-Review · NeurIPS 2019]

As the title suggests, this paper uses learning-theoretic tools to study a problem of estimating a (Lipschitz) spatial field using sensors which are location-unaware. The main contributions are the formulation of the sensing problem, a proposed algorithm, an analysis of its sample complexity, and some proof-of concept experiments. This work may be a solid start towards an interesting (new) class of problems which are amenable to probabilistic/learning-theoretic techniques. While in the future this may involve new "contributions to statistical learning theory," the present study does not really develop new techniques. Overall, the problem is interesting but the paper could be strengthened significantly in several directions as noted by the reviewers. In particular some more specific motivating examples would help ground the paper -- the authors mention "spatial sensing... in smart cities or IoT or climatology" but do not elaborate. This would help the readers better evaluate the appropriateness of various mathematical assumptions. The goal of the authors is to present a more abstract formulation, but designing practical schemes may only be possible under more restrictive assumptions. The (asymptotic) analysis of the algorithm seems to be the first step towards a more complete story which could provide results for the finite sample setting. The authors allude to this in their response, and reviewers felt that the current manuscript would have been much stronger with such results. As it stands, the $N$ chosen for experiments is quite small (as the authors have it, "shy of infinity") so it is unclear to what degree the experiments are reflective of the theoretical contributions.